# Giant Juvenile Fibroadenoma: Case Report and Review of the Literature

**DOI:** 10.3390/jcm12051855

**Published:** 2023-02-26

**Authors:** Anna Eleftheriades, Ermioni Tsarna, Konstantina Toutoudaki, Eleni Paschalidou, Nikolaos Christopoulos, Ioannis Georgopoulos, Georgia Mitropoulou, Panagiotis Christopoulos

**Affiliations:** 1Postgraduate Programme “Maternal Fetal Medicine”, Medical School, National and Kapodistrian University of Athens, 11527 Athens, Greece; 2Second Department of Obstetrics and Gynecology, “Aretaieion” Hospital, Faculty of Medicine, National and Kapodistrian University of Athens, 11528 Athens, Greece; 3Department of Pediatric Surgery, “Agia Sofia” Children’s Hospital, 11527 Athens, Greece; 4Department of Pathology, “Agia Sofia” Children’s Hospital, 11527 Athens, Greece

**Keywords:** juvenile fibroadenoma, giant fibroadenoma, adolescence, breast disease

## Abstract

Fibroadenomas are common benign breast tumors. Fibroadenomas that exceed 5 cm in diameter, weigh more than 500 g, or replace more than four-fifths of the breast are characterized as giant. A fibroadenoma diagnosed in patients during childhood or adolescence is characterized as juvenile. An extensive PubMed search of the literature in English up until August 2022 was performed. In addition, a rare case of a gigantic fibroadenoma in an 11-year-old premenarchal girl who was referred to our adolescent gynecology center is presented here. Eighty-seven cases of giant juvenile fibroadenomas have been reported in the literature along with our case. Patients with giant juvenile fibroadenoma presented at a mean age of 13.92 years and usually after menarche. Juvenile fibroadenomas are usually unilateral, occurring either in the right or the left breast; the majority of them are diagnosed when they are already more than 10 cm in size, and they are most frequently treated with total lump excision. Differential diagnosis includes phyllodes tumors and pseudo-angiomatous stromal hyperplasia. Conservative management is feasible, but surgical excision is recommended to patients with suspicious imaging features or when the mass grows rapidly.

## 1. Introduction

Fibroadenomas are common breast tumors in children and adolescents, even though breast disease is exceptionally rare among pediatric and adolescent patients, and its nature is highly different in comparison to adults. Fibroadenomas represent 30–50% of palpable breast masses during childhood and adolescence and 44–94% of surgically excised breast masses in the same age group [1,2]. A fibroadenoma can be characterized as gigantic or giant when it has a diameter exceeding 5 cm, weighs more than 500 g, or replaces more than four-fifths of the breast [1,2]. Giant fibroadenomas have been reported among minors, even though they represent a rare clinical entity. When a fibroadenoma is diagnosed in patients between 10 and 18 years old, it is characterized as juvenile fibroadenoma [3]. A broader definition of juvenile fibroadenoma includes all cases in childhood and adolescence, thus up to 19 years old based on the definition of adolescence by the World Health Organization (WHO) [4].

Fibroadenomas are benign masses that consist of estrogen-sensitive epithelial and stromal tissue. Malignancy in a pre-existing juvenile fibroadenoma is very rare [5]. Fibroadenomas are not linked to an increased risk of breast cancer in the general population. However, in women with a significant family history of breast cancer and in cases of proliferative changes in the histopathologic examination of the mass, an increase in breast cancer risk has been reported [6]. Overall, breast cancers in adolescents are extremely rare, accounting for 0.1% of all breast cancers in women regardless of age and less than 1% of all pediatric cancers [1].

## 2. Materials and Methods

In order to identify reported cases of giant juvenile fibroadenomas, we performed an extensive PubMed search of the literature in English, up until August 2022. A list of keywords including “giant fibroadenoma”, “giant juvenile fibroadenoma”, “fibroadenoma in adolescence”, and “juvenile fibroadenoma” was used in our search algorithm. Case reports, case series, cohorts, and randomized control trials were eligible for our review. For the purpose of this review, juvenile characterized all of the cases that were diagnosed up to the age of 19 years, based on the WHO definition for adolescence [4]. Reports were excluded if the full text was not accessible, if the paper was published in a language other than English, or if the age criterion (≤19 years old) could not be confirmed for all cases based on the data provided by the authors of the report. The following details were recorded for each case: country of origin, size and location of the tumor, age of the patient, menarcheal status, and type of treatment. The reference lists from all reviewed papers were screened for papers potentially eligible for our review. In addition, in this paper, we describe a novel case of an 11-year-old premenarcheal patient who developed a giant fibroadenoma during adolescence. 

## 3. Results

To the best of our knowledge, 87 cases of giant juvenile fibroadenomas have been reported in the literature along with the case presented in this review [7,8,9,10,11,12,13,14,15,16,17,18,19,20,21,22,23,24,25,26,27,28,29,30,31,32,33,34,35,36,37,38,39,40,41,42,43,44,45,46,47,48,49,50,51,52,53,54,55,56,57,58,59,60,61,62,63,64,65,66]. The mean age of the patients was 13.92 years. Unfortunately, in 44/87 (50.6%) cases, the menarcheal status of the patient was not reported. From the remaining patients, 12 (13.8%) were premenarcheal and 31 (35.6%) were postmenarcheal. Of the total, 72/87 cases (82.8%) of fibroadenomas occurred unilaterally and 15/87 cases (17.2 %) occurred bilaterally. Among the 72 cases with unilateral fibroadenomas, 25 (34.7%) were located in the right breast and 23 (31.9%) in the left breast, whereas the location of the mass was not reported for 24 patients (33.3 %). With regard to their size, in 17/87 cases (19.5%) the maximum diameter of the largest fibroadenoma was below 10 cm, and in 48/87 cases (55.2%) it was at least 10 cm, whereas such information was not provided for 22 (25.3%) cases. The cases for which maximum diameter was not reported were characterized as giant juvenile fibroadenomas, because they weighed more than 500 g or replaced more than four-fifths of the breast. Notably, in 12 (13.8%) cases, the fibroadenoma was at least 20 cm is size. With regard to the applied treatment, 71/87 (81.6%) patients underwent lump excision, 7/87 (8%) patients underwent mastectomy, in 4/87 (4.6%) patients, endoscopic techniques were applied, one patient had to undergo lump excision from the right breast and synchronous mastectomy of the left breast, and in the remaining four patients, other surgical techniques were applied or details were not reported. In summary, based on our literature review, giant juvenile fibroadenomas are usually unilateral, occurring either in the right or left breast, the majority of them are diagnosed when they are already at least 10 cm in size, and they are most frequently treated with total lump excision. Patients with giant juvenile fibroadenoma present at a mean age of 13.92 years and usually after menarche, although caution should be exercised as menarcheal status was not reported for half of the reviewed cases.

The case of an 11-year-old girl, who presented in our clinic with a palpable mass in her right breast, is also included in this review. The mass was identified during self-palpation 6 months before presentation, as it increased rapidly in size. However, no other complaints such as pain, fever, dyspnea, cough, or loss of appetite were reported. With regard to her medical history, she reported a tonsillectomy and eustachian tube surgery at the age of three and a half years old. Moreover, the patient had been diagnosed with pelviureteric junction stenosis at the age of 9 years. As far as family history is concerned, her mother reported a non-giant fibroadenoma excision at the age of 28, as well as surgical excision of ovarian endometriomas. 

Upon clinical examination, an asymmetry of the breasts was evident, with the right breast having greater size than the left one (Figure 1). However, no skin discoloration or discharge from the nipples was observed. During palpation, a firm mass with discrete margins in the outer half of the right breast was detected. No axillary lymph nodes were palpated. Residual systems’ examination was unremarkable.

Ultrasound examination of the breast showed a hypoechoic mass in the outer half of the right breast. Its maximum diameter measured 10 cm, with significant internal vascularity. Breast MRI indicated a firm mass, with clear margins and characteristics of a benign mass, most probably a giant fibroadenoma. Its dimensions were calculated as 7.4 cm × 6 cm × 8.1 cm (Figure 2).

Subsequently, the patient was admitted to a pediatric surgical clinic, where she underwent a total surgical excision of the mass (Figure 3). Reconstructive surgery was not required, since no visible asymmetry was observed after the excision (Figure 3).

The histopathologic evaluation of the surgical specimen confirmed the suspected diagnosis of giant juvenile fibroadenoma. The macroscopic examination revealed a mass with smooth margins. Its dimensions were 10 cm × 8 cm × 2.5 cm (Figure 4). To be noted, the dimensions of the fibroadenoma differed significantly from the ones calculated during the breast MRI and were rather closer to the ultrasonographic estimation. The microscopic examination revealed fibroepithelial composition of the neoplasm. Immunohistochemistry indicated expression of actin and CD34, confirming the stromal origin of one component of the tumor. Simultaneously, epithelial tissue tested positive for estrogen and progesterone receptors, b-catenin and e-cadherin. Ki-67/MIB-1 cellular proliferation markers were detected among 5–10% of fibrous cells. During the follow-up consultation, breast development was normal and symmetrical. In addition, the patient reported satisfaction with the aesthetic result. Interestingly, the patient’s period started one month after the operation. Table 1 presents the 87 cases of giant juvenile fibroadenomas which were included in our review [7,8,9,10,11,12,13,14,15,16,17,18,19,20,21,22,23,24,25,26,27,28,29,30,31,32,33,34,35,36,37,38,39,40,41,42,43,44,45,46,47,48,49,50,51,52,53,54,55,56,57,58,59,60,61,62,63,64,65,66].

## 4. Discussion

The most common breast tumors in children and adolescents are fibroadenomas, which are benign tumors with glandular and stromal components. Typical clinical findings include a painful, slowly enlarging mass that may cause breast enlargement or asymmetry. Clinical examination reveals a usually smooth, firm, non-tender mass that is mobile during palpation. Nonetheless, a fibroadenoma might occasionally be painless and present as a rapidly growing mass, as in the case of our patient. With regard to age at diagnosis of giant juvenile fibroadenomas, Michael Sosin et al., in a systematic literature review based on 153 patients, reported a mean age of 16.7 years; among these cases, the mean lesion diameter was 11.2 cm [7]. Of note, Sosin et al. included in their systematic review cases diagnosed up to the age of 25 years, therefore, including cases of giant fibroadenomas after adolescence. As a result, the mean age reported by Sosin et al. at diagnosis was higher than it would have been if cases of giant fibroadenomas after adolescence were excluded from the analysis. 

The precise etiology of juvenile fibroadenomas remains unknown. Reproductive hormones could play a role, since estrogen and progesterone receptors are expressed in fibroadenomas, and these lesions occur more frequently in puberty, pregnancy, and in individuals who take oral contraceptives [67]. Genetic predisposition also plays a role, since fibroadenomas appear more commonly in African-American females, and occasionally patients report positive family history of breast fibroadenomas, as was the case in our patient [8]. 

With regard to imaging, fibroadenomas appear as hypoechoic during ultrasonography and show diffuse enhancement on magnetic resonance imaging (MRI). Typical diagnostic workup includes patient history, physical examination, and radiographic evaluation with ultrasonography being the reference standard technique [7]. A study by Smith et al. showed that in 78.8% of fibroadenoma cases, histopathology confirmed the diagnosis made through ultrasound, whereas color Doppler revealed non-vascularization or minimal internal vascularity in most cases [68]. Nonetheless, the imaging of internal vascularity, as in our patient, cannot exclude fibroadenoma from the differential diagnosis. Magnetic resonance imaging (MRI), even though it is not the first method of choice, can also be a helpful tool in the evaluation of the exact location of the lesion. Mammography is not performed routinely, as the risk of breast malignancy is low, and the adolescent breast tissue is radiographically dense, hindering mammography’s diagnostic value [19]. 

The differential diagnoses in cases of giant juvenile fibroadenomas include phyllodes tumors and pseudoangiomatous stromal hyperplasia. Of note, the majority of fibroadenomas naturally grow slowly and eventually regress before they reach 5 cm in diameter. Fibroadenomas in adolescents are not considered as a premalignant state, even though some reports have suggested the possibility of rare malignant transformation, and metachronous lesions are frequently reported [19]. In particular, the risk of malignant degeneration of a juvenile fibroadenoma is reported to be less than 0.3%, and the most common fibroadenoma-related malignancy is lobular malignant tumor [30,69]. Histopathologically, proliferative changes, such as sclerosing adenosis, duct epithelial hyperplasia, epithelial classification, and papillary apocrine changes are thought to be indicative of increased breast cancer risk [70]. 

Many authors suggest that pediatric and adolescent patients with fibroadenomas, presenting typical findings, both clinical and ultrasonographic, can be managed conservatively rather than surgically. The surgical excision of a fibroadenoma can cause scarring at the incision site, dimpling of the breast, duct system damage, and ultimately mammographic post-surgical changes that lower the diagnostic accuracy of mammography later in life. In addition, a recurrence is expected in 10–25% of the patients [71]. In cases managed conservatively, a safe management option is to regularly follow up the mass with ultrasound in order to confirm the stability of the lesion over time. For those who desire a definite confirmation of the diagnosis, fine-needle aspiration or core needle biopsy can be offered [7]. However, if the surgical excision of the mass has been decided, a core needle biopsy or other invasive testing methods prior to the operation should be avoided, since pediatric patients do not easily tolerate invasive procedures that may negatively impact them psychologically and emotionally. In our case, core biopsy was not performed since surgical resection was the optimal treatment method upon diagnosis, due to the rapid growth pattern of the mass and the presence of internal vascularity in ultrasound, and an additional invasive procedure, such as core biopsy, would not alter the surgical plan for this patient.

With regard to surgical management of juvenile fibroadenomas, the indications include suspicious imaging characteristics or rapid growth. In fact, juvenile fibroadenomas, especially giant, are commonly treated with excision because of their rapid growth in contrast to typical fibroadenomas that appear in adulthood [7]. In the majority of giant juvenile fibroadenoma cases, lump excision is surgically feasible. Mastectomy is performed in very rare cases of recurrent giant fibroadenomas, and surgeons will have to take into consideration the potential morbidity, both physical and psychological [7]. Mammoplasty should be performed after the mastectomy to minimize the psychological trauma. Prosthesis or autologous tissue reconstruction may be used [41]. Another possible treatment method that has been described in the literature is cryoablation, which can be performed after core needle biopsy: however, the available data refer only to adult fibroadenomas [72,73,74]. 

Juvenile fibroadenomas are a separate and distinguished clinical entity in comparison to adult fibroadenomas, and their management may differ. As mentioned above, reports of malignant degeneration of a juvenile fibroadenoma are extremely rare [5]. Adult fibroadenomas, however, have been considered as a long-term risk factor for the development of breast cancer. A study published in the New England Journal of Medicine showed that the risk of invasive breast cancer was 2.17 times higher in women with fibroadenomas in comparison to the controls (CI 95%M 1.5 to 3.2), and this risk was found to be higher in patients with complex fibroadenomas, proliferative disease or family history of the disease [6,75]. This possible association is also supported by a clinicopathologic study of women with carcinomas that developed within fibroadenomas, where the mean age of such patients was higher in comparison to patients with benign fibroadenomas [75]. For this reason, a mammography is also recommended to older women to exclude cancerous lesions. In mammography, fibroadenomas appear as well-circumscribed, homogeneous nodules, which may include inner calcifications [75]. Because of the increased incidence of carcinoma in older women, most clinicians recommend the excision of persistent breast masses, such as fibroadenomas, as patients approach adulthood [76]. In younger patients, conservative management with frequent clinical evaluations can be more safely performed; in fact, several investigators suggest that the cut-off age should be 25 [75]. Even though the breast cancer risk has been well described in the literature to differ between fibroadenomas in adolescence and adulthood, the underlying cause has not been explored. One could argue that the aforementioned findings might reflect the well-known increasing risk of breast cancer with age. On the other hand, it is possible that distinct genetic profiles of juvenile versus adult fibroadenomas contribute to the aforementioned difference in breast cancer risk.

The management of giant fibroadenomas also differs from that of simple fibroadenomas. Giant fibroadenomas appear to be more cellular and have less lobular components in comparison to simple fibroadenomas [75]. Since they are larger in size, they can compress normal breast tissue, causing greater symptoms of discomfort and pain, and therefore, they are more commonly excised. Another reason for this is that giant fibroadenomas cannot be easily distinguished from phyllodes tumors through physical examination or ultrasound [77]. On the other hand, management of simple fibroadenomas entails careful follow-up, since most simple fibroadenomas in adolescents decrease in size and may even disappear with time [78]. Despite the aforementioned differences between giant and simple fibroadenomas in adolescence that refer mostly to clinical presentation and management, it remains unclear whether distinct pathogenetic mechanisms are involved in giant juvenile fibroadenomas as opposed to simple fibroadenomas in adolescence.

Juvenile fibroadenomas are a rare clinical entity, and there is a lack of case series as well as clear guidelines regarding the diagnostic evaluation and treatment options. Therefore, research data arise mostly from case reports that do not consistently report clinical and histopathological data. The purpose of our literature review was to evaluate the reported cases of juvenile giant fibroadenomas. To the best of our knowledge, this is the largest and most current clinical review of giant juvenile fibroadenoma and the only one in which a strict age criterion was applied, leading to the exclusion of giant fibroadenoma cases after adolescence. This review, however, included only level four and five evidence, namely case reports and case series, which may be a factor of bias, including publication bias. In our review, we identified that menarcheal status is frequently not reported, even though this would be important, as reproductive hormones are thought to participate in fibroadenoma pathogenesis. If the majority of patients with juvenile fibroadenomas were premenarcheal, that would indicate that other factors contribute more to the pathogenesis of fibroadenomas in adolescence. Similarly, details regarding reproductive hormone levels and reproductive hormone receptors in the surgical specimens were under-reported in the reviewed case reports. In order to shed light on the pathogenesis of juvenile fibroadenomas and develop evidence-based clinical guidelines for diagnosis, management, and treatment of this rare entity, more robust data are needed.

## 5. Conclusions

Enlarging breast masses in children and adolescents can be of great concern to young patients, and fibroadenomas should be included in the differential diagnosis. Generally, the management of breast disease in children and adolescents is different from that in adults. Special surgical consideration needs to be placed on maintaining lactation ability, preserving the developing breast parenchyma through minimal injury of the breast, and achieving good cosmetic results because of the patients’ young age. A further consultation with a plastic surgery specialist may also be required, especially in those cases when breast symmetry is not easily achieved after a standard lump excision. Patients need to receive appropriate counseling and to be closely followed up for future breast masses or abnormalities that may form as they progress through adolescence and adulthood. In summary and based on our review, the authors conclude that ultrasonography is the most common method of evaluation and total lump excision is the most common treatment option. Therefore, ultrasonography is recommended as an initial diagnostic tool. However, the currently available data do not suffice for the purpose of proposing standard recommendations for adolescent patients presenting with such breast lesions.

## Figures and Tables

**Figure 1 jcm-12-01855-f001:**
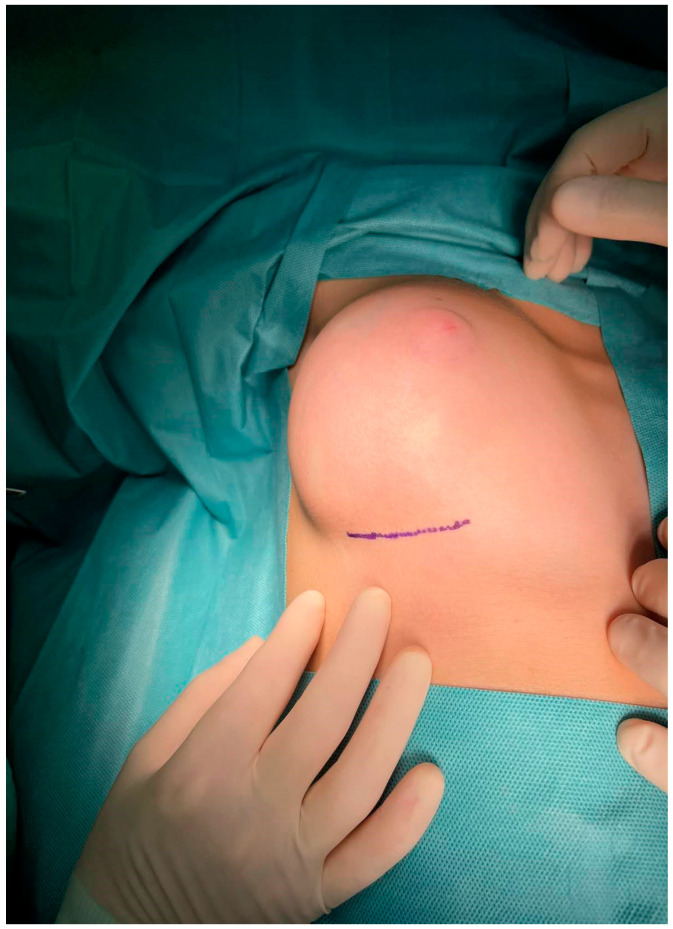
Macroscopic view of the right breast of the 11-year-old patient before surgery.

**Figure 2 jcm-12-01855-f002:**
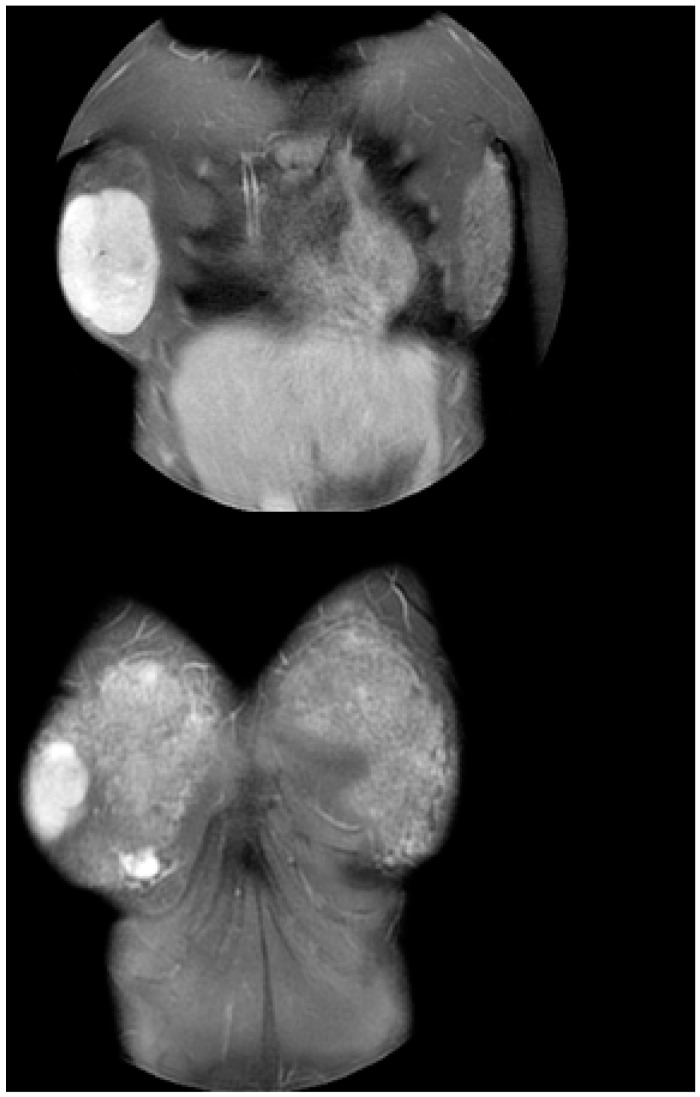
MRI imaging of a 7.4 cm × 6 cm × 8.1 cm mass with clear margins and characteristics compatible with giant fibroadenoma.

**Figure 3 jcm-12-01855-f003:**
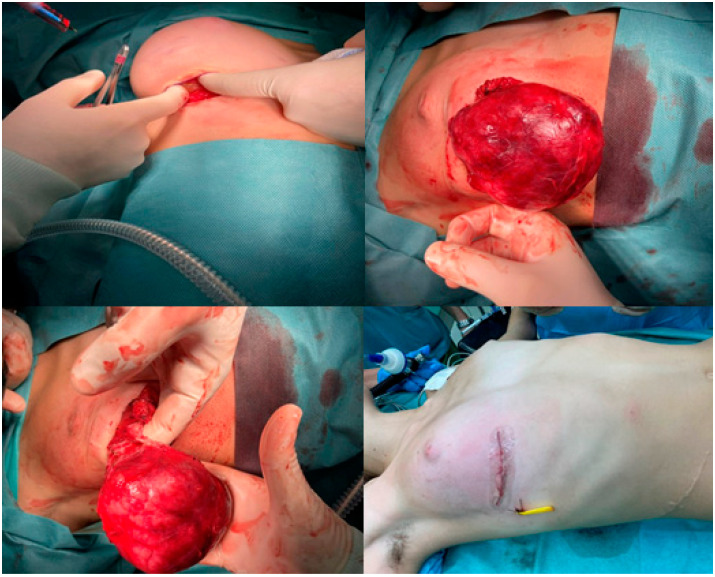
Total surgical excision of the fibroadenoma without reconstructive surgery.

**Figure 4 jcm-12-01855-f004:**
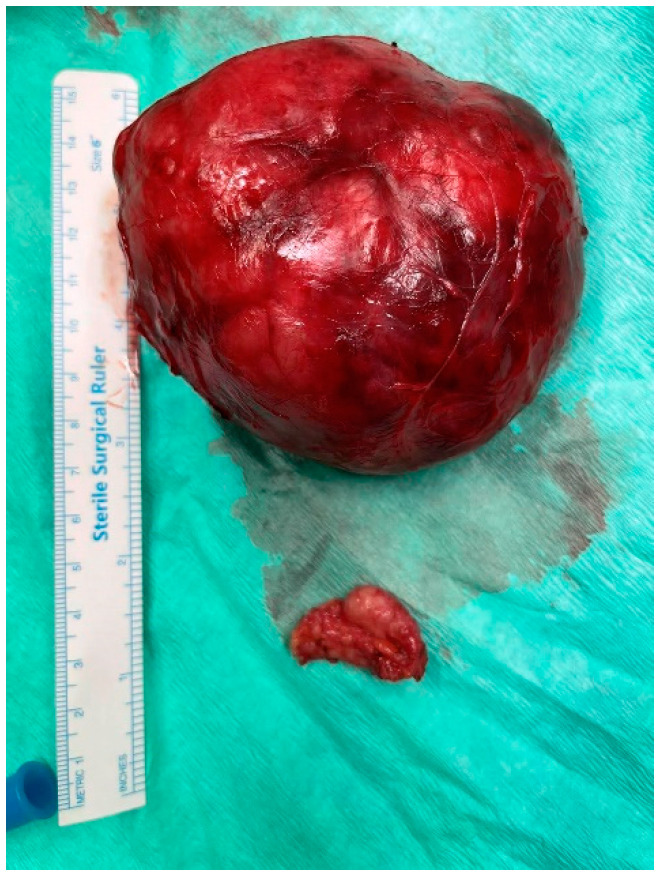
Surgical specimen of the excised mass with dimensions 10 cm × 8 cm × 2.5 cm.

**Table 1 jcm-12-01855-t001:** Characteristics of the cases of juvenile fibroadenomas, identified through an extensive PubMed search.

Authors	Country	Size (cm)	Unilateral–Bilateral	Location (Right/Left)	Age (Years)	Menarcheal Status (pre/post)	Treatment	No. of Patients
Celik et al. 2015 [52]	Turkey	9.5 × 8 × 6	unilateral	right	10	N/A	total lump excision	1
Yu et al. 2021 [61]	China	9.65 × 5.80 × 5.5	unilateral	left	11	N/A	single-hole breast endoscopic resection	1
Zeeshan et al. 2021 [62]	Pakistan	13.5 × 11.5 × 6	unilateral	left	13	post	total lump excision	1
Baral et al. 2020 [63]	Nepal	9 × 7	unilateral	right	18	post	total lump excision	1
Islam et al. 2019 [64]	Trinidad and Tobago	28 × 25	unilateral	left	16	post	total lump excision	1
Makkar et al. 2017 [9]	India	right lump: 6 × 5/ left lumps: 10 × 6 and 2 × 3	bilateral	N/A	14	pre	bilateral total lump excisions	1
Celik S. et al. 2017 [10]	India	17 × 13 × 5.5	unilateral	left	14	post	total lump excision	1
Jategaonkar et al. 2018 [11]	India	right lump: 63 × 47/ left lump: 51 × 39	bilateral	N/A	17	N/A	bilateral simple mastectomy with synchronous breast reconstructions	1
Juarez et al. 2021 [12]	Mexico	16 × 13	unilateral	left	14	N/A	total lump excision	1
Kupsik et al. 2017 [13]	USA	11.2 × 8.5 × 6.7	unilateral	right	9	pre	total lump excision	1
Khan et al. 2015 [14]	Pakistan	right lump: 6 × 5/ left lump: 10 × 15	bilateral	N/A	10	pre	bilateral total lump excisions	1
Giannos et al. 2017 [19]	Greece	14 × 12	unilateral	right	12	pre	total lump excision	1
Gaurav et al. 2015 [15]	India	20 × 20	unilateral	left	10	pre	total lump excision	1
Mohd Firdaus et al. 2017 [16]	Malaysia	12 × 9 × 5	unilateral	right	12	N/A	total lump excision	1
Gkali et al. 2017 [17]	Greece	15 × 13	unilateral	right	12	pre	total lump excision	1
Nishtha et al. 2021 [18]	India	15.5 × 9 × 10.5	unilateral	right	14	pre	total lump excision	1
Bayramoglu et al. 2018 [20]	Turkey	8.3 × 9.9 × 5.5	unilateral	left	14	N/A	partial mastectomy with synchronous breast reconstruction with an implant	1
Rafeek et al. 2016 [21]	India	right lump: 39 × 22 × 17/ left lump: 40 × 24 × 18	bilateral	N/A	13	post	bilateral total lump excisions with synchronous reduction mammoplasty	1
Kitazawa et al. 2022 [22]	Japan	13 × 9 × 4.5	unilateral	left	14	N/A	total lump excision	1
Wang et al. 2020 [23]	China	6 × 6	unilateral	left	19	N/A	total lump excision	1
Heng et al. 2020 [24]	Malaysia	10.6 × 14.5 × 15.1	unilateral	right	19	post	mastectomy with synchronous breast reconstruction	1
Tian et al. 2020 [59]	China	22 × 16 × 11	unilateral	right	18	N/A	total lump excision	1
Kaur et al. 2015 [30]	India	numerous tumors on both breasts, number not specified, size 1.5–2.5	bilateral	N/A	15	post	bilateral subcutaneous mastectomy	1
Kozomara et al. 2016 [41]	Boznia Herzegovina	20 × 20 × 15	unilateral	left	16	N/A	reduction mammoplasty technique of the left breast	1
Ezer et al. 2013 [25]	Turkey	P1: 25 × 12/ P2: 30 × 16/ P3: 45 × 20/ P4: 60 × 35	unilateral	not specified	P1: 14, P2: 17, P3: 10, P4: 14	N/A	total lump excision	4
Matz et al. 2013 [26]	USA	8.9 × 9.8 × 8.8	unilateral	left	13	N/A	total lump excision	1
Arowolo et al. 2013 [27]	Nigeria	20 × 30	unilateral	left	14	pre	total lump excision with synchronous breast reconstruction	1
Biswas et al. 2012 [28]	Bangladesh	P1: 15 × 13/ P2: 11 × 11	unilateral	P1: right, P2: left	P1: 14, P2: 16	N/A	total lump excision	2
Sosin et al. 2012 [7]	USA	12.1	unilateral	right	13	N/A	total lump excision	1
Cheng et al. 2012 [29]	USA	P1: 5/ P2: 8/ P3: 10.5	unilateral	P1: left, P2: left, P3: left	P1: 17, P2: 13, P3: 15	N/A	endoscopic specimen pouch technique	3
Heilmann et al. 2012 [31]	Germany	13 × 12.5 × 4.5	unilateral	right	17	post	total lump excision	1
Izadpanah et al. 2012 [32]	Canada	7.3 × 9.3 × 8.9	unilateral	left	12	post	total lump excision	1
Chepla et al. 2011 [34]	USA	P1: 11.6 × 9.8 × 11.3/ P2: 15 × 4 × 4	unilateral	P1: left/P2: right	P1: 10, P2: 16	N/A	total lump excision	2
Ng et al. 2011 [35]	Canada	19	unilateral	right	17	N/A	total lump excision	1
Nikumbh et al. 2011 [36]	India	right lump: 15 × 12/ left lump: 17 × 15	bilateral	N/A	12	pre	total excision of bilateral breast lumps conserving the normal breast tissue, nipple, and areola by the plastic surgeon	1
Tantrige et al. 2011 [54]	UK	12	unilateral	left	13	N/A	total lump excision	1
Yagnik et al. 2011 [37]	India	13 × 11 × 12	unilateral	right	15	N/A	total lump excision	1
Poh et al. 2010 [38]	USA	not described	bilateral	N/A	12	post	Right breast: excision of 6 discreet masses through vertical incision, laterally based pedicle used for nipple–areola complex/Left breast: subcutaneous mastectomy through “inverted T” incision, nipple–areola complex was salvaged as a free nipple graft.	1
Biggers et al. 2009 [39]	USA	P1: 5.9/ P2: 10.5/ P3: 12/ P4: 17	unilateral	not specified	P1: 11/ P2: 12/ P3: 14/ P4: 15	N/A	total lump excision (inframammary approach)	4
Calcaterra et al. 2009 [40]	Italy	14 × 16 × 17	unilateral	right	12	N/A	total lump excision (inframammary approach)	1
Gobbi et al. 2009 [42]	Italy	P1: 8/ P2: 10	unilateral	P1: left/ P2: left	P1: 12/ P2: 15	P1: N/A/ P2: post	total lump excision	2
Mukhopadhyay et al. 2009 [43]	India	right lump: 22 × 20/ left lump: 18 × 16	bilateral	N/A	11	pre	bilateral lump excision conserving normal breast tissue, nipple, and areola	1
Wolfram et al. 2009 [44]	Austria	5	unilateral	right	15	N/A	total lump excision	1
Dolmans et al. 2007 [45]	Netherlands	9 × 9	unilateral	left	18	N/A	nipple-sparing subcutaneous mastectomy of the left breast	1
Moore et al. 2007 [46]	USA	right breast fibrofatty mass (1040 g): 17.0 × 15.0 × 7.0/ left breast fibrofatty mass (1111 g): 18.5 × 16 × 7.5	bilateral	N/A	9	pre	bilateral subtotal mastectomies	1
Ahuja et al. 2005 [47]	India	not described	unilateral	right	12	N/A	total lump excision	1
Lee et al. 2004 [48]	South Korea	multiple nodules, the largest: 17 × 3 × 3, others not described	bilateral	N/A	11	post	central pedicle reduction mammoplasty	1
Zacharia et al. 2003 [49]	India	8 × 5	unilateral	right	13	N/A	total lump excision	1
Hanna et al. 2002 [50]	Kuwait	right lump: 6 × 6 × 5/ left lump: 5.5 × 5 × 6	bilateral	N/A	15	N/A	not specified	1
Wechselberger et al. 2002 [51]	Austria	not described	unilateral	right	15	N/A	total lump excision	1
Davis et al. 2001 [53]	USA	5 × 3.5 × 2.5	unilateral	right	19	N/A (CAIS)	total lump excision	1
Baxi et al. 2000 [55]	India	right breast: 17 firm nodules (mean diameter 4, range 1–8)/ left breast: 9 (mean diameter 8, range 1–11)	bilateral	N/A	16	post	total lump excision	1
Kamei et al. 2000 [56]	Japan	P1: 11 × 10 × 5/ P2: 10 × 9 × 8	unilateral	left	P1: 19/ P2: 17	N/A	total lump excision	2
Simmons et al. 2000 [57]	USA	16	unilateral	right	12	N/A	total lump excision	1
Silfen et al. 1999 [58]	South Africa	not described	bilateral	N/A	13	N/A	bilateral reduction mammoplasty	1
Hoffman et al. 1978 [60]	US	not described	unilateral	right	13	N/A	total lump excision with synchronous reduction mammoplasty	1
Song et al. 2014 [8]	South Korea	13 × 8	unilateral	right	12	post	total lump excision	1
Mashiloane 2000 [65]	South Africa	not described	unilateral	right	16	post (during pregnancy)	mastectomy	1
Ugburo et al. 2012 [66]	Nigeria	not described	14 patients unilateral, 2 patients bilateral	not specified	mean 14.06 ± 0.42 (range 12–18)	post	total lump excision	16
Eleftheriades et al. 2022 (our case)	Greece	10 × 8 × 2.5	unilateral	right	11	pre	total lump excision	1
							Total Number of Cases:	87

Abbreviations: N/A, not available; P, patient; CAIS, complete androgen insensitivity syndrome.

## Data Availability

The authors confirm that the data supporting the findings of this study are available within the article.

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
