# Peer review of "Giant Juvenile Fibroadenoma: Case Report and Review of the Literature"

_jcm, 2023, doi:10.3390/jcm12051855_

Round 1
Reviewer 1 Report
Fibroadenomas are common breast tumors in children and adolescents, even though breast disease is exceptionally rare among pediatric and adolescent patients, and its nature is highly different in comparison to adults. Fibroadenomas represent 30-50% of palpable breast masses during childhood and adolescence and 44-94% of surgically excised breast masses at the same age group. Genetic predisposition also plays a role, since fibroadenomas appear more commonly in African-American females and occasionally patients report positive family history for breast fibroadenomas, as was the case of our patient.
The title and content of the article represent a topic of real interest worldwide. At the level of the specialized literature, there are few articles regarding juvenile fibroadenoma, having an absolute heterogeneity. In what constitutes the establishment of diagnostic and treatment standards, no consensus was found.Reproductive hormones could play a role since estrogen and progesterone receptors are expressed in fibroadenomas and these lesions occur more frequently in puberty, pregnancy, and in individuals who take oral contraceptives.
The introduction of the article presents originality by proposing a topic with a huge academic potential.
The bibliographic data inserted along the article presents a qualitative chronology. The subject of the article represents a true scientific revolution in its field.
The material and methods section of the article presents a quantitative and qualitative exposition of the research plan, respectively a good reproducibility in order to develop other studies with this theme. I consider it necessary to develop new studies on this subject and implement them on a population scale.
The results of the article present a logical and chronological exposition outlining a rare pathology for which there is no absolute consensus regarding the methods of diagnosis and treatment. The figures and tables keep a specific chronology throughout their exposition, presenting qualitative aspects related to the subject of the article.
The topic of the article is a real interest for the future with major importance in this field. I consider it necessary to develop new studies on this subject and implement them on a population scale. The article presents an important research point with an optimal linguistic exposition, having an exponential potential for the future. This present article is written in a clear and concise manner.
The article presents originality, with an optimal literary exposition, representing a topic of real interest for the future with objective results at the research level.
Reviewer 2 Report
Giant juvenile fibroadenoma: Case Report and Review of the Literature by Eleftheriades et al.
GENERAL COMMENT
The authors present a review article, which lacks a figure showing the flow diagram.
The methodology is deficient, only one database was consulted (Pub Med) and it is suggested to adhere to the PRISMA method.
Additionally, they show a clinical case of a patient, which confuses the reader. Being a review, this information is not in accordance with the type of article.
It is suggested to the authors to look for more databases of the information of interest and the clinical case, to present it separately in another article and later, to resubmit it to MDPI.
Reviewer 3 Report
Giant fibroadenomas are fibroadenomas larger than 5 cm, or 500 grams in size, they are rare benign breast lesions that account for approximately 0.5%–2% of fibroadenomas. They usually occur in pregnant or lactating women or adolescent women. Eleftheriades A, et al. submitted a review of a giant juvenile fibroadenoma together with a case report. The manuscript summarized the characteristics of the juvenile fibroadenoma published, which include patient’s age, tumor size, location, treatment, and menarche status, etc. The authors also reported a case of giant fibroadenoma.
This manuscript contributes to the field. However, the biggest issue of this paper is language problem. Grammatical errors exist in many places in the entire manuscript.
Here are accouple of examples in the abstract.
in line 16, Giant is a singular, why use “are”?
in line 23-24, the authors stated that “Juvenile fibroadenomas are usually unilateral, occurring both at the right and the left breast,” Since it usually occurs on one side,why use “both…and” it should be “either… or”.
In line 27-28, it might be better if it is edited to the following: Conservative treatment is feasible, but a surgical resection is highly recommended to the patients with suspicious imaging features, or when the mass grows rapidly.
The author needs to proofread the manuscript carefully before submitting.
In line 70, “/” in front of 44/87 (50.6%) cases is an extra, which should be deleted.
In line 72, “In 72/87 cases (82.8%) the fibroadenoma was unilateral and in 5/87 cases (17.2 %) bilateral.
The authors should know what the subject is. Obviously, fibroadenomas is subject. 72/87 cases (82.8%) of fibroadenomas occurred unilaterally. In addition, there is no predicate verb “in 5/87 cases (17.2 %) bilateral”. It should be “5/87 cases (17.2%) occurred bilaterally”.
In line 90, “In this review, is included a case or an 11-year…” it should be “a case of an 11-year-old girl is included” ……
There are too many places in the paper that need to be edited or modified.
Reviewer 4 Report
As the authors mention, Juvenile Fibroadenoma is a rare entity and the present manuscript represents the largest series of cases reviewed. Therefore, it could contribute valuable information regarding these rare tumors. However, there are weaknesses and points that are not clear and would be necessary, in my opinion, for the authors to provide/clarify.
It is not always clear whether the authors refer to just Juvenile Fibroadenoma or the giant variant.
For example, the title suggests the review refers specifically to the giant variant (also stated at the bottom of p. 10). However, 25% of the cases did not have diameter information.
The basis for using 19 years as a limit for “juvenile” should be discussed.
It would be interesting if the authors could include other data regarding the patient’s mother’s Fibroadenoma: age, was it giant?
It would be valuable if the authors could conclude in this reviewn whether there any unique feature of Giant Fibroadenomas (other than the diameter) compared to Fibroadenomas in general?
What are the differences between Juvenile Fibroadenomas and adult Fibroadenomas (other than the age)? For example, is there a difference in family history between the two groups?
The last 2 points in the criticism are important, since the manuscript as is does not really provide conclusions on features that would be distinct ones for this subset of tumors.
Round 2
Reviewer 2 Report
Manuscript title: Giant juvenile fibroadenoma: Case Report and Review of the Literature
Manuscript ID: jcm-2177912
The authors responded to the suggestions made.
The bibliography was updated and the content improved.
I suggest revising the English.
Reviewer 3 Report
This manuscript has been greatly improved after revision.
However, an abbreviation of WHO should be added in after "World Health Organization" in line 42. Otherwise, WHO in line 58 should be listed the full name.
Reviewer 4 Report
As the authors mention, Juvenile Fibroadenoma is a rare entity and the present manuscript represents the largest series of cases reviewed. Therefore, it could contribute valuable information regarding these rare tumors. However, there are weaknesses and points that are not clear and would be necessary, in my opinion, for the authors to provide/clarify.
It is not always clear whether the authors refer to just Juvenile Fibroadenoma or the giant variant.
For example, the title suggests the review refers specifically to the giant variant (also stated at the bottom of p. 10). However, 25% of the cases did not have diameter information.
The basis for using 19 years as a limit for “juvenile” should be discussed.
It would be interesting if the authors could include other data regarding the patient’s mother’s Fibroadenoma: age, was it giant?
It would be valuable if the authors could conclude in this reviewn whether there any unique feature of Giant Fibroadenomas (other than the diameter) compared to Fibroadenomas in general?
What are the differences between Juvenile Fibroadenomas and adult Fibroadenomas (other than the age)? For example, is there a difference in family history between the two groups?
The last 2 points in the criticism are important, since the manuscript as is does not really provide conclusions on features that would be distinct ones for this subset of tumors.
